# Enhanced Catalytic Effect of Ti_2_CT_x_-MXene on Thermal Decomposition Behavior of Ammonium Perchlorate

**DOI:** 10.3390/ma16010344

**Published:** 2022-12-30

**Authors:** Jingxiao Li, Yulei Du, Xiaoyong Wang, Xuge Zhi

**Affiliations:** 1School of Mechanical Engineering, Nanjing Vocational University of Industry of Technology, Nanjing 210023, China; 2School of Mechanical Engineering, Nanjing University of Science and Technology, Nanjing 210094, China

**Keywords:** MXene, ammonium perchlorate (AP), thermal decomposition, catalyze

## Abstract

Transition metal carbonitrides (MXenes) are promising catalysts due to their special structures. Recently, many studies have shown that MXenes have a catalytic effect on the thermal decomposition of ammonium perchlorate (AP). However, the catalytic effects have not been extensively investigated. Therefore, it is important to illustrate the catalytic mechanisms of pure MXene in AP thermal decomposition. Herein, the catalytic properties of Ti_2_CT_x_ for ammonium perchlorate (AP) thermal decomposition were investigated by numerous catalytic experiments. The results showed that the high-temperature decomposition (HTD) decreased by 83 °C, and the decomposition heat of AP mixed with Ti_2_CT_x_ increased by 1897.3 J/g. Moreover, the mass spectrum (MS) data showed that the NH_3_, H_2_O, O_2_, N_2_O, NO, HCl, and NO_2_ were formed. In addition, according to the X-ray diffraction (XRD), Raman spectrum, high-resolution transmission electron microscopy (HRTEM), selected area electron diffraction (SAED), and X-ray photoelectron spectra (XPS) results, the Ti_2_CT_x_ nanosheets can adsorb the gaseous products and react with them in-situ, generating anatase-TiO_2_ and carbon layers. The Ti_2_CT_x_, as-resulted anatase-TiO_2_, and carbon can synergize and further catalyze the thermal decomposition of AP when both electron and proton transfers are accelerated during AP decomposition.

## 1. Introduction

MXenes, a new group of two-dimensional (2D) early transition metal carbonitrides, are synthesized from the etching of the A layers from M_n+1_AX_n_ phases. In this formula, M stands for an early transition metal; A stands for A-group elements; X stands for carbonitrides; and n is 1, 2, or 3 [1,2]. Notably, with the termination of O, OH, and/or F groups on the surface of MXene, M_n+1_X_n_T_x_ becomes more accurate to describe the MXene, where T_x_ stands for the surface terminations [2]. Due to its good electrical conductivity, large inner surface area, abundant surface terminations, and graphene-like morphology, MXene has been increasingly investigated as a potential material for sensors [3], lithium-ion batteries [4,5], supercapacitors [6,7], metal adsorption materials [8], catalysts [9,10], etc.

Ammonium perchlorate (NH_4_ClO_4_, AP) is one of the important oxidative and energetic materials in solid rocket propellants, which provides a powerful thrust to rockets by releasing a large amount of heat and gases [11]. Therefore, to improve the performance of AP thermal decomposition, various catalysts including metals [12], metal oxides [13,14], carbon or carbon-based materials [15,16], and so on, have been exploited for many decades. Recently, due to their large surface area, special 2D layered structure, and excellent electrical conductivity, MXenes also have been used directly as catalysts or ideal substrates to support nanoparticles in boosting the thermal decomposition or burning of AP [17,18,19,20,21]. For instance, by high-temperature heating, Gao and his co-workers prepared MXene-Cu_2_O composites [18]. Afterward, the as-resulted hybrids were found to greatly influence the thermal decomposition of AP, decreasing its HTD temperature to 121.4 °C. Similarly, Li et al. [19] have reported MXene as a matrix for constructing MXene/MnCo_2_O_4.5_ composites with 3D flower-like structures by a hydrothermal process. Simultaneously, MXene/MnCo_2_O_4.5_ composites also showed high catalytic activity for AP decomposition. Alternatively, the surface of MXene nanosheets mentioned later could be oxidized during the high-temperature preparation process. Therefore, to prevent MXene from being oxidized, Tan and co-workers [20] fabricated MXene/NiO composites through in-situ precipitation at room temperature and calcination in N_2_. The compound was highly active in the thermal decomposition of AP, giving an HTD temperature reduction value of 50.6 °C, which was higher than that of NiO alone. Likewise, to protect the surface metal atoms of MXene from oxidation, Zhu and co-workers coated carbonaceous materials on the surface of MXene to construct MXene@C first before the preparation of MnO_2_/MXene [21]. These MnO_2_/MXene@C hybrids have greatly accelerated the thermal decomposition of AP, reducing the high-temperature decomposition by 128.8 °C. These studies have shown that adding MXene and MXene-based materials can facilitate the thermal decomposition of AP. However, the effects of MXene on the catalyzation of AP decomposition have not been extensively investigated.

MXenes are sensitive to gases owing to their 2D special structures, numerous highly active functional sites, hydrophilic nature, and high surface area. For example, based on first-principle calculations, Ti_2_CT_x_ was used to test the adsorption behaviors of gases, such as NH_3_, H_2_, CH_4_, CO, CO_2_, N_2_, NO_2,_ and O_2_ [22]. The result revealed that NH_3_ can be chemisorbed, and the others are physisorbed on the Ti_2_CT_x_ sheet because of a different adsorption energy. Similarly, other MXenes, such as Ti_3_C_2_, Sc_2_CO_2_, and Ti_2_NS_2,_ also can be confirmed to have great potential applications in gas storage by calculating the adsorption energies of gases on them [23,24,25,26]. These adsorption behaviors can affect the progress of MXenes’ catalytic reaction on AP, because gaseous products, such as NH_3_, N_2_, NO, NO_2_, O_2_, N_2_O, H_2_O, Cl_2_, HClO_4_, etc., would be generated during AP degradation [27,28,29]. In addition, owing to numerous exposed metal atoms on the surface, MXenes lack resistance to the oxidizing gases above, resulting in an oxidation reaction and generation of transition metal oxides (TMOs)/MXene composites and carbon. Therefore, MXenes involve a series of complex catalytic reaction processes. Based on this, it is imperative to investigate the catalytic mechanisms of pure MXene in AP thermal decomposition.

Compared to multilayered MXenes, their delaminated ones often exhibit better properties [30,31]. However, the delamination of multilayered MXenes into separated flakes requires intercalation steps after the A layers’ etching process. Furthermore, the intercalated species can have a permanent influence on the MXenes, which could introduce additional factors for investigation on the catalytic mechanism. In this work, in order to avoid the introduced factors, a multilayered Ti_2_CT_x_ MXene was selected to catalyze AP thermal decomposition. Then, the catalytic mechanism of Ti_2_CT_x_ on the degradation of AP was studied.

## 2. Materials and Methods

The preparation of Ti_2_CT_x_ powders has been reported in detail in our previous work [32]. Briefly, the precursor phase Ti_2_AlC was exposed to aqueous HF (40 wt.%) for 2.5 h at room temperature; then, the as-prepared suspension was cleaned by DI water several times; at last, purified samples were freeze-dried to obtain Ti_2_CT_x_ flakes. The AP particles were brought from Guoyao and preheated to 60 °C for 24 h in a vacuum oven to remove moisture. Then, a series of weight ratios of Ti_2_CT_x_ (0, 5 wt.%, 10 wt.%, 20 wt.%, 30 wt.%) were added into AP, accordingly, and ground in an agate mortar for 15 min.

The surface structure, composition, and morphology of samples were analyzed by scanning electron microscope (SEM, Quant 250F, FEI, Hillsboro, OR, USA), X-ray diffraction (XRD, Bruker-AXS D8 Advance, Bruker, Leipzig, Germany) with Cu-Kα radiation ( λ = 1.54178 Å) operated at 40 mA and 40 kV), transmission electron microscope (TEM, JEM-200CX, JEOL, Tokyo, Japan) operated at 200 kV, X-ray photoelectron spectroscopy (XPS, PHI 5000 VersaProbe, Ulvac Phi, Chigasaki, Japan, Al-Kα radiation), and Raman spectroscopy (JY HR800, JY, France, λ = 514 nm).

The catalytic effect of the prepared samples on the thermal decomposition of AP was studied by using an STA449F TG-DSC at heating rates of 5, 10, 20 °C/min, and between 40 and 500 °C. The purge gas used was 99.995% pure Argon. The overall purge flow rate was maintained at 20 mL/min. Simultaneously, a Netzsch (Aeolos QMS403, Selb, Germany) mass spectrometer was used to study the decomposition products.

## 3. Results and Discussion

### 3.1. Thermal Decomposition of Pure AP

The morphologies and structures of AP and Ti_2_CT_x_ were characterized by SEM and XRD, as shown in Figure 1. Figure 1a shows that the AP particles were relatively coarse with various sizes, and in Figure 1b, all diffraction peaks were ascribed to AP, confirming its purity. Figure 1c shows the typical 2D stacked multilayered structure of Ti_2_CT_x_. Moreover, the XRD pattern in Figure 1d shows that the diffraction peaks were located at around 9°, which also confirmed the formation of Ti_2_CT_x_ [33].

Figure 1e shows the TG and DSC curves of pure AP, which reveal that the thermal decomposition of pure AP occurred in three stages. In the first stage, the endothermic peak appeared at 242 °C without weight loss, which is due to the transition from orthorhombic to cubic form [34]. In the second stage (named low-temperature decomposition, LTD), the exothermic peak at 311 °C was assigned to the partial decomposition of AP with 17.5 wt.% weight loss, and some intermediates were formed by dissociation and sublimation. The third peak (high-temperature decomposition, HTD) appeared at a relatively higher temperature of 447 °C, indicating the complete decomposition of the intermediate to volatile products.

For describing the release of volatiles, MS was used to follow the representative mass fragments of the substances produced by AP. As shown in Figure 1f, the evolution profiles of NH_2_^+^ (*m*/*z* = 16), O^+^ (*m*/*z* = 16), NH_3_^+^ (*m*/*z* = 17), H_2_O (*m*/*z* = 18), NO (*m*/*z* = 30), O_2_ (*m*/*z* = 32), HCl (*m*/*z* = 36), N_2_O (*m*/*z* = 44), and NO_2_^+^(*m*/*z* = 46) along with temperatures were presented with 10 °C min^−1^ by MS. The curves above show that the main decomposition products in LTD were NH_3_, H_2_O, O_2,_ and N_2_O, indicating that AP was first decomposed into NH_3_ and HClO_4_, and HClO_4_ was further decomposed into oxidizing substances, and then these oxidizing substances reacted with NH_3_, forming H_2_O, O_2,_ and N_2_O. Compared with the LTD stage, the ion current intensity in the HTD stage was higher, suggesting that the decomposition of AP was mainly concentrated in this stage. The broad peak type of each ion flow indicates that the decomposition reaction of pure AP proceeded slowly, in which the NH_3_, H_2_O, O_2_, N_2_O, NO, HCl, and NO_2_ were mainly examined. In addition, the formation temperature of HCl complied with that of H_2_O, but later than other products, which illustrates that HCl is a secondary product.

### 3.2. The Catalytic Activity of Ti_2_CT_x_

Figure 2 shows that the catalytic performance of various ratios’ addition of Ti_2_CT_x_ on thermal decomposition of AP was tested by DSC and TG. Figure 2a shows the DSC curves of AP in the presence of 0, 5 wt.%, 10 wt.%, 20 wt.%, and 30 wt.% Ti_2_CT_x_. As shown, the results indicated that Ti_2_CT_x_ did not affect the crystal transformation. The changes in peak decomposition temperature on DSC curves and initial decomposition temperature on TG curves (in Figure 2b) show the positively catalytic role of Ti_2_CT_x_ in the thermal decomposition of AP. With increasing Ti_2_CT_x_ content, the LTD peak rarely changed, but two exothermic peaks appeared in the HTD stage, and both exothermic peaks moved towards the temperature reduction direction, and finally became a sharp exothermic peak when the additional amount of Ti_2_CT_x_ was up to 30 wt.%. The HTD temperatures of AP-*x* wt.% Ti_2_CT_x_ (*x* = 5, 10, 20, and 30) mixtures were 435, 415, 385, and 359 °C, respectively, and decreased by 12, 32, 63, and 88 °C compared with pure AP (447 °C), respectively. In addition, the TG curves show that the addition of Ti_2_CT_x_ affected the initial decomposition temperature of AP, decreasing it by 15 °C when adding 5~20 wt.% Ti_2_CT_x_ and 35 °C when adding 30 wt.% Ti_2_CT_x_, respectively. These results suggest that Ti_2_CT_x_ has a catalytic effect on the decomposition of both LTD and HTD.

In Figure 2c, the released energies of the mixtures were 1103, 1332, 2087, and 2527 J/g corresponding to AP-*x* wt.% Ti_2_CT_x_ (*x* = 5, 10, 20, 30) mixtures, enhanced by 473.3, 702.3, 1457.3, and 1897.3 J/g compared to 629.7 J/g of pure AP. However, it is obvious that the growth rates of released energies were not always positive, which is probably due to the excessive addition of Ti_2_CT_x_-MXene. In addition, as shown in Table 1, MXene-based composites exhibited better catalytic activity when the mixture ratio of the catalyst was low.

Figure 3 demonstrates the MS results of AP in the presence of Ti_2_CT_x_ (0, 10, and 30 wt.%). AP was completely decomposed at lower temperatures and shorter times. In addition, the results show that the intensity of the ion currents of NO, NO_2_, H_2_O, and HCl of AP mixed with Ti_2_CT_x_ was lower than that of pure AP, suggesting that the layered Ti_2_CT_x_ easily absorbed the gaseous products. Notably, the intensity of the ion currents of O_2_ and N_2_O of Ap-10 wt.% Ti_2_CT_x_ was higher than that of pure AP, which indicates that the addition of Ti_2_CT_x_ promoted the formation of the two gases. However, when 30 wt.% Ti_2_CT_x_ was added, the change trend of the MS results of O_2_ and N_2_O complied with that of other ions. This also complies with the previous report [22,24] that Ti_2_CT_x_ with special structures can adsorb gases and be easily oxidized by oxidizing gases. Therefore, it can be inferred that Ti_2_CT_x_ could absorb O_2_ and N_2_O and react with the two gases. Usually, this phenomenon occurs in the thermal decomposition process of AP catalyzed by transition metals, and the generated transition metal oxides will synergistically catalyze the thermal decomposition of AP [37].

### 3.3. Catalytic Mechanism of Ti_2_CT_x_

To reveal the thermal decomposition mechanism of AP catalyzed by MXene, the Ti_2_CT_x_ after catalysis was characterized by the SEM, XRD, Raman Spectroscope, XPS, TEM, and HRTEM.

Figure 4a shows the XRD results of the Ti_2_CT_x_ after the catalytic thermal decomposition of AP. After catalyzing the thermal decomposition of AP from room temperature to 500 °C, peak centers at 25.3°, 36.9°, 37.8°, 48°, 53.9°, 55°, 62.7°, 68.8°, 70°, 75°, and 76° appeared in all the samples, which can respectively be attributed to (101), (103), (004), (200), (105), (211), (204), (116), (220), (215), and (301) planes of anatase-TiO_2_ (A-TiO_2_) according to JCPDS 21-1272. The very weak peak at 27.48° belonged to rutile-TiO_2_ (JCPDS 21-1276), which demonstrates that a little amount of rutile-TiO_2_ was formed. Three extra peaks located at 35.9°, 42.7°, and 60.5° can be indexed to TiC (JCPDS 65-0242), which were the inevitable intermediate products during Ti_2_AlC preparation, and it was hard to remove. Notably, the typical peak of Ti_2_CT_x_ shifted to a lower angle. Based on these results, A-TiO_2_ particles were formed during the catalytic process, which paralleled heating MXenes in the air or other atmospheres [38,39]. The formation of A-TiO_2_ was further explained by the Raman results (shown in Figure 4b). After the thermal decomposition catalyzed by AP, four peaks centered at 144, 399, 519, and 639 cm^−1^ appeared, which responded well to the vibrational modes E_g(1)_, B_1g(1)_, A_1g_, and E_g(3)_ of anatase (A-TiO_2_), respectively [40]. Meanwhile, relatively weak D- and G-bands were observed (shown in the inset of Figure 4b), indicating that only a small amount of carbon was formed on the surface of Ti_2_CT_x_ nanosheets with the formation of A-TiO_2_.

Figure 5 shows XPS patterns of the surface and inner (after 10 s Ar sputtering) of samples after the catalytic thermal decomposition of AP. The fitting components are listed in Table 2. On the sample surfaces, almost 79% of the Ti 2p region belonged to TiO_2_, and 38% of the O 1s region belonged to TiO_2_. However, after 10 s Ar sputtering, the percentage of the Ti 2p region and O 1s region decreased to 3% and 7%, respectively. This suggested that the Ti atoms on the surface of the Ti_2_CT_x_ nanosheets were oxidized to TiO_2_ during the process of accelerating the exothermic decomposition of AP. The interior structures of the Ti_2_CT_x_ were still maintained. These results demonstrated that the C-Ti bonds were broken, forming TiO_2_ particles and carbon layers. Additionally, there were still numerous C-Ti-T_x_ bonds [41], and the carbon layers stuck the TiO_2_ particles on the surface of Ti_2_CT_x_ like “glue”, which was helpful to maintain the stability of Ti_2_CT_x_.

The SEM image (Figure 6a,b) shows that the interlamellar spacing of Ti_2_CT_x_ became larger after the catalytic thermal decomposition of AP. This amplification was partially due to A-TiO_2_ formation on the external surface, and presumably also between the layers. However, no typical A-TiO_2_ particles were observed from the SEM image. Therefore, further confirmation was conducted by the TEM analysis. The TEM image (Figure 6c) and HRTEM image (Figure 6d) demonstrated that some Ti_2_CT_x_ nanosheets still retained their planar structure, while others transformed into small crystalline particles. The calculated *d*-spacing values of the latter phase agreed with values for anatase in the JCPDS card (21-1272). The indexed SAED pattern (shown in Figure 6e), taken from the square region, combined two phases: the original Ti_2_CT_x_ and anatase-TiO_2_. Herein, it is also clear that Ti_2_CT_x_ underwent surface oxidation when the catalytic exothermic decomposition of AP happened.

Generally, there are two main viewpoints on the mechanism of the thermal decomposition of AP (NH_4_ClO_4_)–electron transfer and proton transfer–and the process is usually as follows:(1)electrons (e^−^) transfer from ClO_4_^−^ to NH_4_^+^:
ClO_4_^−^ NH_4_^+^ → ClO_4_ + NH_4_
(1)

(2)protons (H^+^) transfer from NH_4_^+^ to ClO_4_^−^:

ClO_4_^−^ NH_4_^+^ →HClO_4_ + NH_3_
(2)

In the actual thermal decomposition process of AP, both of these mechanisms may occur, but the occurrence and degree of occurrence will vary under different induced environments. According to the previous results, it can be considered that the mechanism of Ti_2_CT_x_ catalyzed the thermal decomposition of AP in the following aspects.

First, XPS results show that the Ti element in Ti_2_CT_x_ presented multiple valence states. In Rudloff’s report [51], the catalytic effect of transition metal oxides on the thermal decomposition of AP may be attributed to their multivalent change, which can result in the acceleration of electrons’ migration from ClO_4_^−^ to NH_4_^+^ during the redox process, thus boosting the entire decomposition reaction. Here, Ti atoms on the surface of Ti_2_CT_x_ are transition metals, and their 3d orbits are electron-unsaturated, where the empty obits could provide convenient bridges for electron transfer. In addition, Ti^3+^ ions possess separate d electrons in the valence band, which could provide more convenient tunnels for electron movement. Herein, the interaction between Ti_2_CT_x_ and AP could contribute to thermal decomposition, forming NH_3_ and HClO_4_.
NH_4_^+^ ClO_4_^−^ → NH_3_(a) + HClO_4_(a) → NH_3_(g) + HClO_4_(g) (3)

Second, the special layered structure endows Ti_2_CT_x_ with good gas adsorption performance. Based on the first-principle calculation, Yu and co-workers [22] found that the Ti_2_CO_2_ (T_x_ = O_2_) can selectively adsorb NH_3_(g), and the process is reversible, wherein the NH_3_(g) could be desorbed after the strain of Ti_2_CO_2_ releasing. Accordingly, when the NH_3_(g) is formed during the decomposition of AP, it can easily be adsorbed by Ti_2_CT_x_ nanosheets. Thus, the MS results confirmed that no NH_3_(g) signal was detected when Ti_2_CT_x_ nanosheets were added. Accordingly, the adsorption behavior of Ti_2_CT_x_ nanosheets can reduce the concentration of decomposition products, promote thermal decomposition, and provide a place for the oxidation of NH_3_. Moreover, the MS result further confirmed the adsorption ability of Ti_2_CT_x_, showing that the concentration of the resulting H_2_O, O_2,_ and N_2_O gases with Ti_2_CT_x_ addition was lower than that of pure AP, which is important to catalyze thermal decomposition.

Third, as can be seen from the XRD and TEM results above, TiO_2_ was formed when oxidizing gases were emitted by AP decomposition, as follows:Ti_2_CT_x_ + O_2_ → TiO_2_+ Carbon + other products (4)
Ti_2_CT_x_ + N_2_O → TiO_2_ + Carbon + other products (5)
Ti_2_CT_x_ + H_2_O → TiO_2_ + Carbon + other products (6)

In addition, the formation of TiO_2_ could reduce the concentration of gaseous products. Generally, carbon layers and Ti^3+^ ions are formed with the formation of anatase-TiO_2_ particles. Meanwhile, these TiO_2_ particles can prevent Ti_2_CT_x_ from further oxidation and participate in the catalytic reaction as transition metal oxides [52].

Fourth, some electrons would be generated from the semiconductor Ti_2_CT_x_ (T_x_ = OH and O) and the resulting anatase-TiO_2_ when they are excited by thermal energy. These excited electrons alongside electrons transferring from NH_4_^+^ to ClO_4_^−^ would interact with HClO_4_ adsorbed on the surface of Ti_2_CT_x_ layers to form superoxide ion O_2_^−^ [53], then oxidize NH_3_ into H_2_O, NO, NO_2_, N_2_O, and so on [54,55]. Importantly, NH_3_ oxidation is a strongly exothermic reaction, thus increasing the exothermic heat of the reaction.
4 NH_3_ + 5 O_2_ = 4 NO + 6 H_2_O ΔH = 905.9 kJ/mol (7)
4 NH_3_ + 7 O_2_ = 4 NO_2_ + 6 H_2_O ΔH = 282.8 kJ/mol (8)
4 NH_3_ + 4 O_2_ = 2 N_2_O + 6 H_2_O ΔH = 1104.9 kJ/mol (9)

Furthermore, MXenes have excellent thermal conductivity [56]; therefore, the heat conduction rate of the whole reaction system would be significantly improved by the Ti_2_CT_x_ addition.

Based on the abovementioned results and analysis, as Figure 7 shows, the thermal decomposition mechanism of AP catalyzed by Ti_2_CT_x_ is described as follows: above all, Ti_2_CT_x_, TiO_2_, and carbon layers with excellent electron conductivity could boost the transfer of electrons (e^−^) from ClO_4_^−^ to NH_4_^+^, promoting decomposition of NH_4_ClO_4_ and the formation of HClO_4_ and NH_3_; meanwhile, the Ti_2_CT_x_ nanosheets can adsorb gaseous products, reduce the concentration of products, and increase the contact chance of HClO_4_ and NH_3_, and shorten the reaction path as well; the formation of TiO_2_ could also reduce the concentration of gaseous products, accelerating the decomposition; then, excited electrons (e^−^) generated from the semiconductor Ti_2_CT_x_ (x = -OH and = O) and resulting TiO_2_ when they are heated, would be transmitted to HClO_4_, the formation of superoxide O_2_^−^, which could oxidize the NH_3_ to produce the final products and release significant amounts of heat; finally, MXene has good thermal conductivity, which can conduct heat quickly and promote a reaction.

## 4. Conclusions

In conclusion, the Ti_2_CT_x_ showed an enhanced catalytic effect on AP thermal decomposition. The thermal decomposition temperature of the composite decreased by 15, 28, 65, and 83 °C corresponding to 5, 10, 20, and 30 wt.% additives, respectively. X-ray diffraction (XRD) and Raman spectroscopy results showed anatase-TiO_2_ formation during the exothermic decomposition process. The high-resolution transmission electron microscopy (HRTEM), selected area electron diffraction (SAED), and XPS results demonstrated that anatase-TiO_2_ were uniformly distributed on the surface of Ti_2_CT_x_. Based on these results, a possible catalytic mechanism for the thermal decomposition of AP catalyzed by Ti_2_CT_x_ was proposed, which would be beneficial for understanding the catalytic mechanism of MXenes and developing efficient catalysts.

## Figures and Tables

**Figure 1 materials-16-00344-f001:**
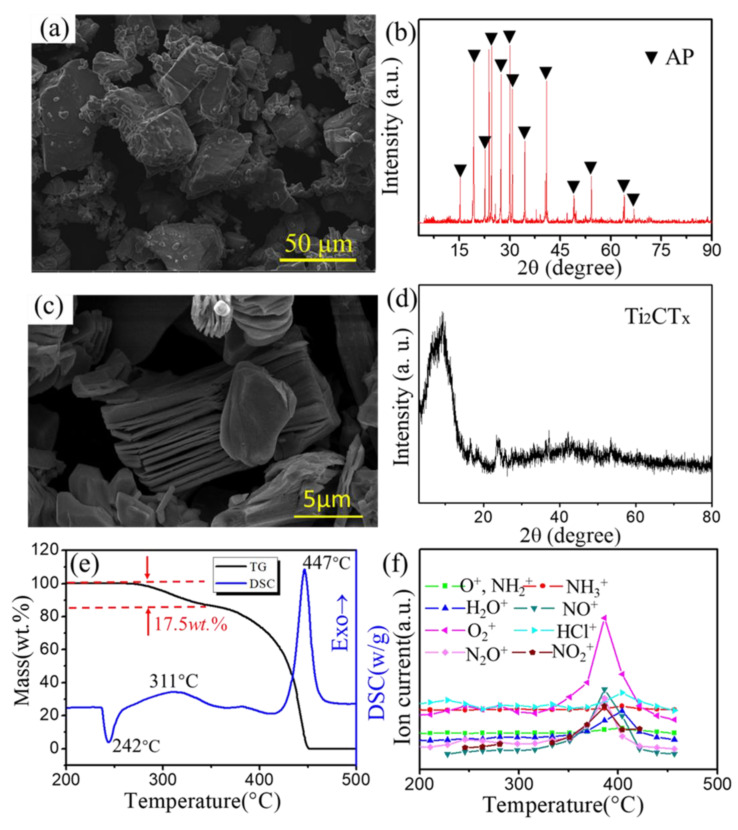
(**a**) SEM image and (**b**) XRD of AP; (**c**) SEM image and (**d**) XRD of Ti_2_CT_x_; (**e**) DSC and TG curves for the thermal decomposition of AP; (**f**) ion intensity curves of pure AP during the thermal decomposition.

**Figure 2 materials-16-00344-f002:**
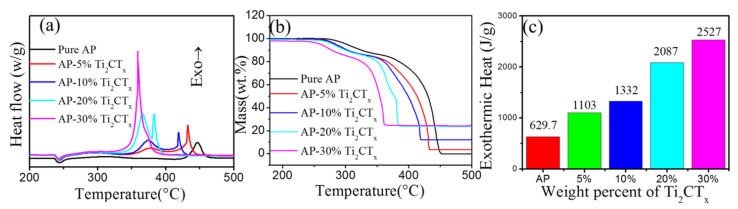
(**a**) DSC, (**b**) TG curves, and (**c**) heat release of pure AP and AP-*x* wt.%Ti_2_CT (*x* = 5, 10, 20, 30 wt.%).

**Figure 3 materials-16-00344-f003:**
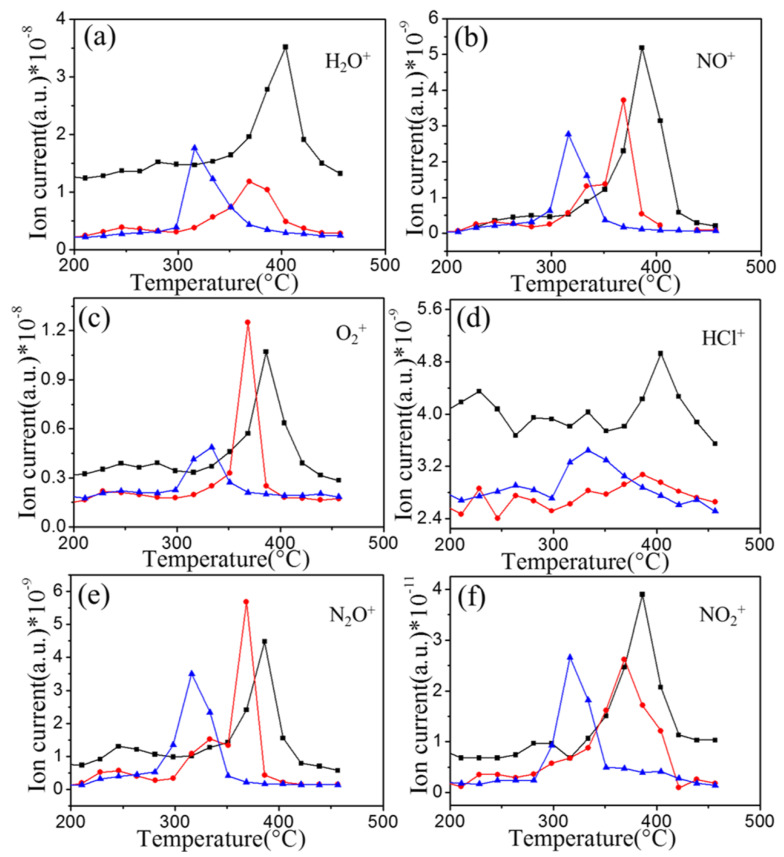
Ion intensity curves of (**a**) H_2_O^+^, (**b**) NO^+^, (**c**) O_2_^+^, (**d**) HCl^+^, (**e**) N_2_O^+^, and (**f**) NO_2_^+^. The black curves, red curves, and blue curves stand for decomposition products of pure AP, AP-10% Ti_2_CT_x_, and AP-30% Ti_2_CT_x_, respectively.

**Figure 4 materials-16-00344-f004:**
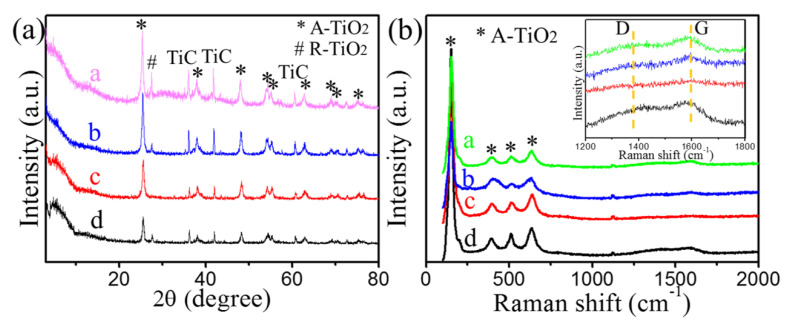
(**a**) XRD patterns and (**b**) Raman spectra of Ti_2_CT_x_ after catalytic thermal decomposition of AP. a, b, c, d stand for Ti_2_CT_x_ after catalytic thermal decomposition of AP in AP-*x* wt.%Ti_2_CT_x_ (*x* = 5, 10, 20, 30 wt.%) samples, respectively. The signs “*” and “#” stand for anatase and rutile, respectively. D and G stand for D-band and G-band of carbon, respectively.

**Figure 5 materials-16-00344-f005:**
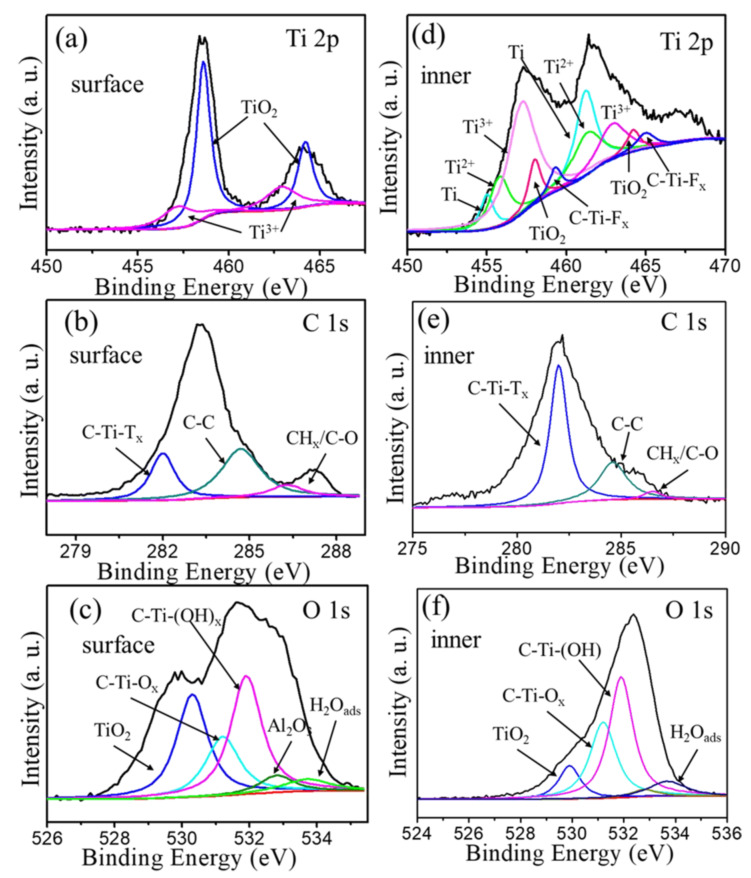
Component peak fitting of XPS spectra for Ti_2_CT_x_ after catalytic thermal decomposition of AP, (**a**) Ti 2p, (**b**) C 1s, and (**c**) O 1s on the surface, and (**d**) Ti 2p, (**e**) C 1s, and (**f**) O 1s in the inner (after 10 s Ar sputtering).

**Figure 6 materials-16-00344-f006:**
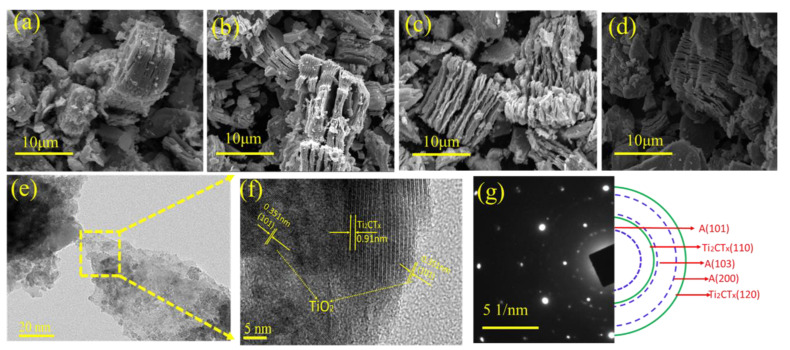
FESEM image of Ti_2_CT_x_ after decomposition of AP (**a**) AP-5 wt.%Ti_2_CT_x_, (**b**) AP-10 wt.%Ti_2_CT_x_, (**c**) AP-20 wt.%Ti_2_CT_x_, and (**d**) AP-30 wt.%Ti_2_CT_x_. (**e**) TEM image of Ti_2_CT_x_ after decomposition of AP and its (**f**) high-resolution TEM image, as well as (**g**) SEAD patterns. “A” stands for anatase-TiO_2_.

**Figure 7 materials-16-00344-f007:**
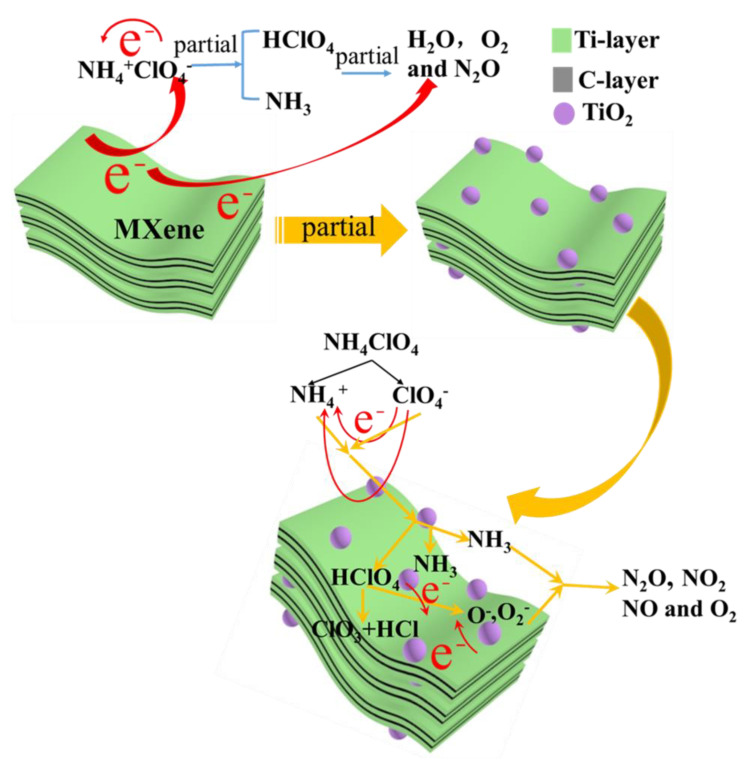
Schematic shows the catalytic mechanism of thermal decomposition of AP by Ti_2_CT_x_.

**Table 1 materials-16-00344-t001:** Comparison of catalytic activity for AP by MXenes and their composites.

Materials	Content (wt.%)	Reduction in HTD Temperature ( °C )	Increase in Decomposition Heat (J/g)	Reference
Ti_2_CT_x_	30	83	1897.3	This work
MXene(Ti_3_C_2_T_x_)	2	12.4	-	[18]
MXene/Cu_2_O		121.4		
MXene/MnCo_2_O_4.5_	4	131.7	1132.4	[19]
MXene(Ti_3_C_2_T_x_)	2	10.2	-	[20]
MXene/NiO		50.6		
MnO_2_/MXene(Ti_3_C_2_T_x_)	2	128.8	-	[21]
MXene (Ti_3_C_2_T_x_)	2	28.1	49.5	[35]
Co_3_O_4_@MXene		108.9	1162.7	
MXene (Ti_3_C_2_T_x_)	2	38.8	73.6	[36]
MXene/ZnCo_2_O_4_		138.3	1079.2	

Note: In this table, HTD represents the high-temperature decomposition.

**Table 2 materials-16-00344-t002:** XPS peak fitting results for Ti_2_CT_x_ after catalyzing AP on the surface and inner locations (after 10 s Ar^+^ sputtering). Numbers in brackets in column 3 are peak locations, and their full widths at half maximum, FWHM, are listed in column 4 in brackets.

Location	Region	BE [eV]	FWHM [eV]	Fraction	Assigned to	Reference
Surface	Ti 2p_3/2_ (2p_1/2_)	457.2 (462.9)	2.1 (2.1)	0.21	Ti^3+^	[42]
		458.6 (464.2)	0.9 (1.0)	0.79	TiO_2_	[43,44]
	O 1s	529.9	1.0	0.38	TiO_2_	[41,44]
		531.2	1.4	0.18	C-Ti-O_x_	[41,45]
		532.0	1.1	0.40	C-Ti-(OH)_x_	[41,45]
		532.8	1.2	0.01	Al_2_O_3_	[45,46,47]
		533.8	2.0	0.03	H_2_O_ads_	[41,45]
	C 1s	282	1.0	0.38	C-Ti-T_x_	[42,48]
		284.7	1.6	0.52	C-C	[49]
		286.3	1.4	0.10	CH_x_/C-O	[49]
Inner (after 10 s sputtering)	Ti 2p_3/2_ (2p_1/2_)	454.8 (461.0)	1.0 (1.9)	0.32	Ti	[42,48]
		455.9 (461.5)	2.2 (2.4)	0.20	Ti^2+^	[42]
		457.5 (463.2)	2.3 (2.0)	0.41	Ti^3+^	[42]
		459.0 (464.7)	1.0 (1.1)	0.03	TiO_2_	[43,44]
		460.4 (466.1)	2.1 (2.9)	0.04	C-Ti-F_x_	[50]
	O 1s	530.3	1.1	0.07	TiO_2_	[41,44]
		531.2	1.2	0.38	C-Ti-O_x_	[41,45]
		531.9	1.1	0.51	C-Ti-(OH)	[41,45]
		533.7	1.7	0.04	H_2_O_ads_	[41,45]
	C 1s	282.0	0.6	0.69	C-Ti-T_x_	[42,48]
		284.6	1.8	0.30	C-C	[49]
		286.5	1.4	0.01	CH_x_/C-O	[49]
	F 1s	685.2	1.8	0.41	C-Ti-F_x_	[50]
		686.2	1.6	0.31	AlF_x_	[46]
		687.3	2.5	0.28	Al(OF)_x_	[46]

## Data Availability

Not applicable.

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
