# Peer review of "Enhanced Catalytic Effect of Ti2CTx-MXene on Thermal Decomposition Behavior of Ammonium Perchlorate"

_materials, 2022, doi:10.3390/ma16010344_

Round 1

Reviewer 1 Report

The catalytic properties of Ti2CTx for ammonium perchlorate (AP) thermal decomposition were investigated and some interesting results are obtained. Recommend publication in Materials after minor revision.

1.       It seems that HTD temperature is important in this study. Please clarify the importance of HTD temperature in abstract in detail.

2.       Fig.1a shows that the AP particles are relatively coarse with sizes around 100 µm. However, it cannot figure out from Fig.a. Based on Fig.a., the particle size must be <100 µm.

3.       “The diffraction peaks are located at around 9°, which also confirmed the formation of Ti2CTx”. Reference is needed here.

4.       HTD decreases with the increase in Ti2CTx content. Why? If so, 50% Ti2CTx maybe much better.

5.       Some spelling mistakes, such as, 60oC, it should be ℃.

6.       HClO4 is further decomposed into oxidizing substances, it should be “HClO4 is further decomposed into oxidizing substances”.

7.       Most References were published more than 10 years ago. Latest references are necessary.

Reviewer 2 Report

The research article 'Enhanced catalytic effect of Ti2CTx-MXene on thermal decomposition behavior of ammonium perchlorate'

In this work, the catalytic properties of Ti2CTx for ammonium perchlorate (AP) thermal decomposition were investigated by numerous catalytic experiments. The results showed that the high-temperature decomposition (HTD) temperature decreased by 83°C and the decomposition heat of AP mixed with Ti2CTx increased by 1897.3 J/g. The experiments are well planned, and this research topic is novel as MXenes now attracts high interest from scientists, researchers and technology experts.

Work requires some improvements:

·         The abstract should be rewritten. The primary importance and demand of this research are not explained, while it is done in the introduction. Suggestion to improve the abstract in a way that would show the importance of this work.

·         What does it mean „tearst“ in the introduction?

·         The introduction lacks information about another type of MXenes application in the thermal decomposition of ammonium perchlorate or other similar substances. Especially such as the most investigated MXenes as Ti3C2Tx. What is the advantages of using Ti2CTx instead of Ti3C2Tx, which is similar in structure?

·         In the introduction part, should be overviewed the size and structure influence (multilayered and delaminated MXenes).

·         The first synthesized, the most common and the most investigated MXene is Ti3C2Tx, not the Ti2CTx.

·         In this study, multilayered MXenes were used as well as in reference [30]. It should be mentioned in MM section

·         Fix typos in the text. Also, in Figure 2C, and Figure 4B.

·         What is the motivation why only 5, 10, 20, 30 30 wt.% of MXenes were tested? Could the Authors add some predictions of the influence of higher wt% of MXenes?

·         This work requires a table with similar investigation results to compare this study’s results.

Do Ti2CTx the best structure among a rich family of MXenes? Would results be better in the case of delaminated MXenes?

Reviewer 3 Report

The submitted manuscript entitled "Enhanced catalytic effect of Ti2CTx-MXene on thermal decomposition behavior of ammonium perchlorate" in Materials is containing really good work and written in a very systematic manner by the authors. However, few minor suggestions are listed below:

1. On page No. 4, the words "exothermal decomposition" should be "exothermic decomposition". Also authors should check it throughout the manuscript.

2. Authors should provide the magnification of each SEM image.

3. Authors should also display the scale bar in Fig. 6 (e).

4. In Fig. 4 (a) and (b), authors should also index the peak after high intensity peaks which is at two theta value ~270 and a low intensity band after wavenumber 1000 should be identify by the authors.

In my view this manuscript needs minor revision and then accepted for publication in Materials. 
